# Biomimetic Gradient Hydrogels with High Toughness and Antibacterial Properties

**DOI:** 10.3390/gels10010006

**Published:** 2023-12-21

**Authors:** Mingzhu Zeng, Zhimao Huang, Xiao Cen, Yinyu Zhao, Fei Xu, Jiru Miao, Quan Zhang, Rong Wang

**Affiliations:** 1Institute of Smart Biomedical Materials, School of Materials Science and Engineering, Zhejiang Sci-Tech University, Hangzhou 310018, China; 2Zhejiang International Scientific and Technological Cooperative Base of Biomedical Materials and Technology, Institute of Biomedical Engineering, Ningbo Institute of Materials Technology and Engineering, Chinese Academy of Sciences, Ningbo 315201, China; 3Ningbo Cixi Institute of Biomedical Engineering, Ningbo 315300, China

**Keywords:** gradient hydrogel, electrophoresis, controlled release, toughness, antibacterial property

## Abstract

Traditional hydrogels, as wound dressings, usually exhibit poor mechanical strength and slow drug release performance in clinical biomedical applications. Although various strategies have been investigated to address the above issues, it remains a challenge to develop a simple method for preparing hydrogels with both toughness and controlled drug release performance. In this study, a tannic acid-reinforced poly (sulfobetaine methacrylate) (TAPS) hydrogel was fabricated via free radical polymerization, and the TAPS hydrogel was subjected to a simple electrophoresis process to obtain the hydrogels with a gradient distribution of copper ions. These gradient hydrogels showed tunable mechanical properties by changing the electrophoresis time. When the electrophoresis time reached 15 min, the hydrogel had a tensile strength of 368.14 kPa, a tensile modulus of 16.17 kPa, and a compressive strength of 42.77 MPa. It could be loaded at 50% compressive strain and then unloaded for up to 70 cycles and maintained a constant compressive stress of 1.50 MPa. The controlled release of copper from different sides of the gradient hydrogels was observed. After 6 h of incubation, the hydrogel exhibited a strong bactericidal effect on Gram-positive *Staphylococcus aureus* and Gram-negative *Escherichia coli*, with low toxicity to NIH/3T3 fibroblasts. The high toughness, controlled release of copper, and enhanced antimicrobial properties of the gradient hydrogels make them excellent candidates for wound dressings in biomedical applications.

## 1. Introduction

The skin is the largest tissue organ in the body and is vulnerable to external damage [1,2]. When the skin is damaged, wound dressings are needed in order to provide a barrier against secondary damage [3,4]. The ideal wound dressing needs to provide a moist microenvironment on the wound surface, absorb wound exudate, and have sustainable antimicrobial activity [5]. Traditional wound dressings, such as gauze and cotton, tend to adhere to the tissue, are not easy to be removed, and have low antimicrobial activity [6]. As wound dressings, hydrogels have received much attention in recent years as they maintain the wound moist and promote autolytic debridement of necrotic tissue [7,8]. Synthetic hydrogels usually have poor mechanical strength, which greatly limit their application in the biomedical field [9,10]. More recently, the development and application of tough hydrogels has become a research hotspot, and various strategies have been explored to develop tough hydrogels [11,12]. Preparation of tough hydrogels can be done via freeze-assisted salting-out, induced crystallization, and induction via radiation of rays [11]. Using a freezing-assisted salting-out fabrication procedure on poly(vinyl alcohol), multi-length-scale hierarchical hydrogels with high strength and toughness were obtained [13]. In addition, researchers have proposed a method to simulate strong hydrogels with muscle contraction by increasing the hydrogen bonding of the polymer chain dispersion medium [14]. However, most strategies to fabricate tough hydrogels are complicated, expensive, time-consuming, and unfriendly to the environment [15].

Gradient functional materials are non-homogeneous functional composites in which two or more materials with different properties are continuously or quasi-continuously varied in one or more dimensional directions via a special preparation process [16]. The advantages of gradient functional materials include their combination of different properties, modulation of the properties according to the environment [17,18]. In nature, many plants and animals developed biological gradient structures to adapt to the changing living environment [19]. For example, the molecular and mechanical gradients of foot filaments allow mussels to adhere to rocks with irregular surfaces [20], the gradient distribution of porous structure in bones directs cell differentiation and osteogenesis [21], and the osmotic/pH/concentration gradients in some plants endow them with reversible motion capabilities [22]. Gradient structure has been adopted in hydrogel design to improve the mechanical properties [23]. For example, Daniele Vigolo et al. demonstrated that the mechanical properties of soft biocompatible materials can be modified via thermophoresis-induced concentration gradients of aqueous sodium alginate solution in microfluidic channels [24]. Strong nanocomposite hydrogels with a designable gradient network structure and mechanical properties were prepared via a facile site-specific photoirradiation strategy [25].

Homogeneous hydrogels have been widely explored as drug delivery systems [26], but usually suffer from burst drug release due to the large difference in drug concentration between the hydrogel matrix and the external environment [27]. The burst release of drugs may result in toxicity and unnecessary waste of therapeutic drugs [28]. Drug release from a gradient hydrogel could be balanced by controlling the difference in drug concentration in the hydrogel matrix, leading to sustainable drug release at the later stage of release [29]. Recently, gradient hydrogels have been increasingly studied to control drug release due to their special biological gradient structure [30,31]. For example, Samad Ahadian et al. developed a novel application of dielectrophoresis to prepare three-dimensional gelatin methacrylate hydrogels with gradients of micro- and nanoparticles containing drugs. This research confirmed that the release of drugs from hydrogels could be achieved in a gradient manner, inducing a cell viability gradient [32]. Although some strategies have been used to prepare gradient hydrogels, there is still a need to develop a simple method for preparing hydrogels with both toughness and controlled drug release performance [33].

Previously, we obtained tannic acid (TA)-reinforced poly (sulfobetaine methacrylate) (polySBMA) hydrogels (TAPS). The TA molecules act as physical cross-linking agents in the polySBMA and as antioxidants and antibacterial agents, endowing them with biological activity [34]. In this study, gradient hydrogels with high mechanical properties, controlled release of antimicrobial agents, and enhanced antimicrobial properties were fabricated via a simple electrophoresis process of the TAPS hydrogels. The electrical field forced Cu ions to diffuse into the hydrogel matrix and formed a gradient cross-linked structure by chelating with the TA molecules and zwitterionic groups. The mechanical properties, antimicrobial properties, antimicrobial agent release profile, and cytotoxicity of the gradient hydrogels were studied to evaluate the application potential of the hydrogel as an antimicrobial wound dressing.

## 2. Results and Discussion

### 2.1. Fabrication of Cu Gradient Hydrogel

First, the TAPS hydrogel was prepared via a straightforward free radical polymerization of SBMA monomers with TA incorporation, as reported in our previous study [34]. The abundant hydrogen bonding and electrostatic interaction between the phenolic hydroxyl groups in TA and the zwitterionic groups in polySBMA formed many physical cross-linking points in the network, which significantly improved the strength and toughness of the hydrogel. Second, the TAPS hydrogel was electrochemically treated to form a gradient structure in the hydrogel (Figure 1a). The hydrogel was subjected to electrophoresis with a Cu plate as the anode and a graphite plate as the cathode. After the power was connected, the color of the hydrogels near the Cu anode gradually changed from light yellow to reddish brown and then to darker yellow-brown. This phenomenon was induced via an electrochemical reaction of the Cu electrode, where the hydrogel could be considered as an electrolytic cell that transformed electrical energy into chemical energy. The current flowed through the electrolyte, resulting in an oxidation reaction on the Cu anode. The standard electrode potential of Cu ions obtained from electron loss is reported to be 0.34 V [35]. In our study, the potential of the Cu electrode was fixed at 5 V, higher than the value required to convert Cu to Cu^2+^. Therefore, an oxidation reaction of Cu occurred on the anode to form Cu^2+^ ions. Cu^2+^ ions were driven into the hydrogel matrix under the applied potential and chelated with the phenolic hydroxyl groups of TA molecules in the hydrogel matrix [36] (Figure 1b). Since the Cu^2+^ ions traveled in the direction from the Cu anode to the graphite cathode [37], they would be complexed with the TA molecules near the Cu plate, then in the central section, and lastly close to the cathode during the electrophoresis process. Therefore, the gradient structure hydrogel cross-linked with Cu ions along the thickness direction (TAPS-CuX) could be obtained. At a fixed voltage of 5 V, the current between the hydrogel decreased from 50–70 mA to around 4 mA within ~1000 s and was maintained at this level afterward. As discussed above, the oxidation of Cu to Cu^2+^ ions occurred due to the electrochemical reaction driven by the applied potential. A proportion of the Cu^2+^ ions that travel in the hydrogel would be chelated by TA molecules. This Cu-TA chelation occurred in a gradient manner along the thickness direction of the hydrogel. In addition, water in the hydrogel was possibly evaporated [38] and electrolyzed [39] as the weight of the TAPS-Cu15 hydrogel decreased by 2.45% after the electrophoresis process. Furthermore, polymer chains and tannic acid molecules in the hydrogel could migrate under the electric field [40]. As a result, the hydrogel near the anode plate first compacted, limiting the transportation of Cu^2+^ ions, and leading to an increase in the resistance of the hydrogel and a decrease in the current (Figure 1a). Since the electrochemical reaction continued to generate Cu^2+^ ions, and some of the chelated Cu^2+^ ions in the hydrogel might escape from the chelating complex under the applied potential, the travel of Cu^2+^ was maintained at a low level, resulting in a low and constant current of ~4 mA within ~1000 s in the electrophoresis process. In sum, the electric charges passing through the hydrogel could be varied by controlling the electrophoretic treatment time (Figure 1b), thus obtaining hydrogels with different gradient cross-linking degrees (i.e., TAPS-Cu0.5, TAPS-Cu1, TAPS-Cu2, TAPS-Cu4, TAPS-Cu7, and TAPS-Cu15). Various parameters and properties of TAPS and TAPS-CuX hydrogels are shown in Table 1.

### 2.2. Mechanical Properties

The mechanical properties of the series of TAPS-CuX hydrogels were investigated. As shown in Figure 2a, the tensile strength of the TAPS hydrogel was measured to be 100.80 kPa. The tensile strength of the hydrogels increased with increasing electrophoresis time, and the maximum value was reached at 368.14 kPa for the TAPS-Cu15 hydrogel. The original TAPS hydrogel was weak with a tensile modulus of 0.85 kPa. The electrophoresis treatment reinforced the hydrogel, and the tensile modulus of the hydrogel increased with the increase in the electrophoresis time (Figure 2b). The value of the tensile modulus of the TAPS-Cu0.5 hydrogel was 1.17 kPa and increased to 5.49 kPa of TAPS-Cu7. Furthermore, increasing the electrophoresis time to 15 min obtained a TAPS-Cu15 hydrogel with a tensile modulus of 16.17 kPa. This outcome could be attributed to the fact that there are a high number of metal chelating bonds between Cu ions and tannic acid molecules (Figure 1b), forming a highly cross-linked structure in the hydrogel, and thereby resulting in stiffening of the hydrogel. Traditional dressings are usually non-stretchable with poor conformity and high modulus compared to skin [41]. The elastic modulus of the TAPS-Cu15 hydrogel falls in the range of human skin (9.5–30.3 kPa) [42], making the TAPS-Cu15 hydrogel less likely to cause tissue contusions when the skin is deformed, and increasing the comfortability and working life of the hydrogel dressing. Therefore, the TAPS-Cu15 hydrogel shows promise as a wound dressing.

Figure 2c shows the compression property of TAPS and TAPS-CuX hydrogels prepared with different electrophoresis times. The compressive strength of the TAPS hydrogel was approximately 20.99 MPa, which is consistent with our previous report [34]. The compressive strength of the hydrogel increased with increasing electrophoresis time and reached a maximum of 42.77 MPa at 15 min of electrophoresis time (that is, the TAPS-Cu15 hydrogel). The compressive strength of the TAPS-Cu15 hydrogel increased approximately 2.04 times compared to that of the TAPS hydrogel (20.99 MPa). It should be noted that the hydrogels were just deformed, not damaged (Appendix A), when compressed at 90% strain. The highest pressure that the skin is subjected to is plantar pressure (maximum of approximately 200 kPa [43]). At 50% compression, the TAPS-Cu15 hydrogel can maintain a pressure of approximately 1.5 MPa, which is sufficient to cope with the in-shoe plantar pressure. Therefore, the TAPS-Cu15 hydrogels were loaded/unloaded for up to 70 cycles at 50% compression strain, and the hydrogels retained excellent integrity, exhibiting a constant compressive stress of 1.50 MPa during the compression cycle (Figure 2d). Therefore, these results confirmed that the tensile and compressive properties of the TAPS hydrogel were enhanced after electrophoresis treatment. This outcome should be attributed to the fact that the Cu ions entered the hydrogel network under the electric field and formed strong metal chelation with the hydroxyl group in the TA molecule, thus enhancing the mechanical properties of the TAPS-CuX hydrogels, and the TAPS-Cu15 was found to be the toughest hydrogel in the study.

### 2.3. Swelling Property

To evaluate the swelling property, the hydrogels were immersed in deionized water to reach the swelling equilibrium (Figure 3a). The swelling ratio of the TAPS hydrogels increased rapidly, ultimately reaching approximately 30% after 1 h. However, the swelling ratio of the TAPS-CuX hydrogels was lower than that of the TAPS hydrogels at 1 h and continued to increase and reached equilibrium after 4 h. The cross-section of the hydrogels after 24 h of swelling was observed using a scanning electron microscope (SEM, regulus 8230, Hitachi, Tokyo, Japan) [44]. The results show that the TAPS-CuX hydrogels exhibited a dense and smooth structure in the cross-section after swelling (Figure 3b). The hydroxyl groups of the TA molecules and the zwitterionic groups of the polySBMA chains in the TAPS hydrogels interact with the water molecules via hydrogen bonding and the solvation effect, causing swelling of the hydrogels. When the TA molecules were chelated with Cu ions, the hydrogen bonding between TA and water molecules was inhibited, resulting in a reduction in the swelling effect of the TAPS-CuX hydrogels.

The swelling property of the hydrogels in artificial sweat was also investigated. As can be seen from Figure 3c, the TAPS hydrogel showed a high swelling ratio of 415.45% in artificial sweat. This is due to the anti-polyelectrolyte effect of zwitterionic polymers, which causes weak interactions between zwitterionic groups in polySBMA chains and results in swelling and extensive hydration in solutions with high ionic strengths [45]. The TAPS-CuX hydrogels showed a significant decrease in equilibrium swelling ratio with the increase in Cu ions in comparison to the TAPS hydrogel. After being soaked in artificial sweat for 24 h, the hydrogel almost reached the equilibrium state, and the equilibrium swelling ratios of the TAPS-CuX hydrogels decreased from 352.86% of TAPS-Cu1 to 307.82% of TAPS-Cu15. The cross-sections of the swelled TAPS and TAPS-CuX hydrogels had distinct wrinkles and build-up of salt deposits in the cross-section (Figure 3d). This is probably due to the shrinkage of the highly swollen hydrogels under the vacuum condition during the freeze-drying process before SEM observation. The swelling results also confirmed that high concentrations of Cu ions in the hydrogel restricted the movement of TA molecules and polySBMA chains via the metal–polyphenol coordination, resulting in an increased degree of cross-linking and a lower swelling ratio of the hydrogel.

### 2.4. Gradient Cu Cross-Linking Structure and Cu Release from the Hydrogel

As discussed above, the content of Cu in the hydrogel can be controlled by varying the electric charge applied in the electrophoresis process, and a gradient distribution of Cu along the thickness direction of the functionalized hydrogel was achieved. The Cu content on the + side, − side, and middle part of the TAPS-Cu15 hydrogel was quantified using inductively coupled plasma-optical emission spectrometry (ICP-OES, SPECTRO ARCOS, SPECTRO, Kleve, Germany) (Figure 4a). As can be seen, the Cu content was 1452.06 μg/g on the + side (contacting the copper anode in the electrophoresis process). It gradually decreased to 1021.99 μg/g in the middle part, and it further decreased to 900.54 μg/g on the − side of the TAPS-Cu15 hydrogel (contacting with the graphite cathode in the electrophoresis process). The Cu content on the + side was 1.42 and 1.61 times of that in the middle part and the − side, respectively. These results confirmed the gradient distribution of Cu content along the thickness direction of the TAPS-CuX hydrogel (Figure 4b).

The Cu release profile from either side of the gradient hydrogel was investigated in an agar-contacting experiment to mimic the conditions of the hydrogel dressing applied on skin tissue. In the test, the TAPS-Cu15 hydrogel was placed with either the + side or the − side contacting the agar gel (Figure 4c). The trend of Cu release from both sides of the hydrogels was similar; that is, Cu released rapidly on the first day, and the release rate was reduced and stable on the following 2 to 5 days (in the range of 2–10 μg/g daily). The higher release rate from the hydrogel on the first day could provide more Cu ions to achieve a stronger sterilization capability (Figure 4d), and the continuous release of Cu ions on the following days could provide a sustainable sterilization effect during the application (Figure 4e). Cu release from the − side (with a lower Cu content) on the first day was 482.93 μg/g, which was lower than that from the + side (that is, 616.36 μg/g). The three-dimensional network structure of the hydrogel itself acts as a Cu reservoir and as a retarding system for the diffusion of Cu ions. The Cu ions distributed a gradient along the thickness direction in the hydrogel matrix. The Cu ions near the − side (containing a lower Cu content) could quickly diffuse out to the agar gel, while the Cu ions near the + side (containing a higher Cu content) need to migrate through the hydrogel matrix before they can diffuse out and reach the agar gel. The TA molecules in the hydrogel matrix would chelate the Cu ions and suppress their migration. In this way, the release behavior of the antimicrobial Cu ions could be controlled, and a more sustainable manner of Cu release was achieved. The controllable release of Cu ions from the gradient hydrogel is beneficial in reducing the potential cytotoxicity of high concentrations of Cu ions to the contacting wound bed, thus achieving a prolonged antimicrobial property of the hydrogel dressing. In addition, the higher Cu content on the + side could endow the outer surface of the hydrogel dressing with a higher contact-killing efficacy against the pathogens from the environment, achieving a strong sterilization effect. Moreover, the hydrogel had a dense structure, and there was no particulate matter found, demonstrating the absence of nanoparticles in the hydrogel. Therefore, we think that copper nanoparticles were not formed in the hydrogel (Appendix A).

### 2.5. Antibacterial Activity

The TAPS hydrogel showed some degree of bactericidal property due to the antibacterial efficacy of TA, as reported in our previous study [34]. It is hypothesized that the antibacterial property could be further enhanced via the introduction of Cu ions in the hydrogel. The antibacterial properties of the hydrogels containing different amounts of Cu ions were evaluated by incubating the hydrogel with Gram-positive *Staphylococcus aureus* (*S. aureus*) and Gram-negative *Escherichia coli* (*E. coli*) (Figure 5). As can be seen, the survival rate of both bacterial species decreased markedly after incubation with the TAPS-CuX hydrogels for 6 h, which indicated that the TAPS-CuX hydrogels had strong antibacterial properties. Specifically, the bactericidal efficacy of the TAPS-Cu0.5 hydrogel against *S. aureus* and *E. coli* was 99.66% and 92.35%, respectively. The bactericidal property of the TAPS-CuX hydrogel increased with increasing Cu content. The TAPS-Cu15 hydrogel almost killed all the bacterial cells of the two strains in the suspension after 6 h of incubation. The TAPS hydrogel is antibacterial due to the addition of TA, possibly because the TA molecule targets peptidoglycans in the cell wall to disrupt bacterial integrity. For the TAPS-CuX hydrogels, the incorporation of Cu would further enhance the antibacterial properties, and its antibacterial capability also depended on the content of Cu in the hydrogel. The positively charged Cu ions released from hydrogels (Figure 4c), reached the negatively charged bacterial cell membrane, adsorbed to the cell membrane via Coulomb gravity, and further penetrated the cell wall, leading to rupture of the cell wall and death of the bacteria [46]. The benefits of using Cu as an antimicrobial agent rather than other metals (such as Ag ions) include the fact that Cu ions are safer compared to other metal antimicrobial agents [47] and that Cu ions are highly stable in light, heat, and aqueous solutions [48]. Furthermore, Cu^2+^ ion is an essential trace element that the body requires to maintain normal hematopoietic function and has an important physiological function in wound healing [49]. According to the literature, the minimum concentration of Cu^2+^ ions with an antibacterial activity of >90% is in the range of 10^−5^ M–10^−4^ M (0.367–3.67 ppm) [50]. The cumulative antimicrobial amount of Cu released from hydrogels over 5 days did not exceed 700 ppm in this study (Figure 4c), which had a good antibacterial effect and low cytotoxicity at this concentration.

### 2.6. Cytotoxicity Assay

The cytotoxicity test of the hydrogels was performed following the ISO-10993-5 standard with minor modification [51]. As shown in Figure 6, the cell viability of all groups with hydrogel extracts was above 90%, indicating that the TAPS-CuX hydrogels had good biocompatibility. As reported in our previous study, TAPS hydrogels have good biocompatibility [34]. On the other hand, Cu is an essential trace element and serves as a bioactive component in vivo. Cu^2+^ has been reported to promote tube formation in vascular endothelial cells by activating the hypoxia-inducible factor 1 pathway [52] and increasing the expression of angiogenesis-related genes, such as vascular endothelial growth factor [53]. Many studies have shown that Cu^2+^ can promote angiogenesis, cell migration, and collagen deposition and thus be effective in tissue regeneration [54], especially skin tissue regeneration. The good biocompatibility and bioactivity of TAPS and Cu indicate the application potential of the TAPS-CuX hydrogel as wound dressings in the biomedical field.

## 3. Conclusions

In this work, a TA-reinforced polySBMA hydrogel (TAPS) was fabricated via free radical polymerization, and the TAPS hydrogel was subjected to a simple electrophoresis process to obtain TAPS-CuX hydrogels with a gradient distribution of Cu ions. The mechanical properties and swelling properties of the TAPS-CuX hydrogels were controlled and optimized by varying the applied electrical charges in the electrophoresis process. The TAPS-Cu15 hydrogel showed good mechanical properties, a skin-like elastic modulus, and low swelling properties, due to the high and gradient cross-linking degree by Cu chelation with TA molecules. The gradient distribution of Cu in the TAPS-Cu15 hydrogel was verified by ICP-OES. The release profile of Cu from the + and − side of the TAPS-Cu15 hydrogel was differentiated due to the gradient distribution of Cu in the hydrogel matrix. It was demonstrated that the initial release rate of Cu from the − side of the hydrogel (containing a lower Cu content) was lower compared to that from the + side (containing a higher Cu content), and the Cu release in the subsequent days was maintained in a low and sustainable level. The Cu release could thus be controlled to obtain a low cytotoxic effect via contact with the − side and a prolonged release profile. The + side of the hydrogel on the outer surface was expected to have a high bactericidal effect against pathogens from the external environment. The TAPS-Cu15 hydrogels were shown to have good antimicrobial properties against Gram-positive *S. aureus* and Gram-negative *E. coli*, and minimal cytotoxicity to mammalian cells. In conclusion, the biomimetic gradient TAPS-CuX hydrogel with good mechanical properties, antimicrobial capacity, minimal cytotoxicity, and controllable and sustainable release of antimicrobials provided a promising solution for the treatment and healing of infected wounds.

## 4. Experimental Section

### 4.1. Materials

Tannic acid, [2-(methacryloyloxy) ethyl] dimethyl-(3-sulfopropyl) ammonium hydroxide (sulfobetaine methacrylate, SBMA, 97%), and ammonium persulfate (APS, 98%) were obtained through Aladdin Chemistry (Shanghai, China). Poly(ethylene glycol) dimethacrylate (PEGDMA, Mn 550 Da) was obtained from Sigma-Aldrich (Shanghai, China). Artificial sweat (Catalogue No.: CF-001) was bought from Chuangfeng Technology (Dongguan, China). *S. aureus* 5622, a strain isolated in patients with skin wound infections, was given by the First Affiliated Hospital of Ningbo University. *E. coli* ATCC 25922 from the American Type Culture Collection. NIH/3T3 fibroblasts cells (Catalogue No.: SCSP-515) were derived from National Collection of Authenticated Cell Cultures, Chinese Academy of Sciences.

### 4.2. Hydrogel Preparation

To a solution of SBMA monomer (11.16 g) in deionized water (9.8 mL), TA (1.845 g), PEGDMA (1.6 mg), and APS (9 mg) were added at room temperature. After degassing with nitrogen bubbles for 30 min, the mixture was poured into a silicone-coated glass mold (12 cm × 7 cm, thickness of 2 mm) and cured overnight at 60 °C to form a hydrogel with a thickness of 2 mm (denoted as TAPS hydrogel). The as-prepared TAPS hydrogel was sandwiched between a Cu plate as the anode and a graphite plate as the cathode to establish a circuit. A power supply (CHI600E, Chinstruments, Shanghai, China) was used with the applied voltage fixed at 5 V. The series TAPS-CuX hydrogels were obtained after the hydrogel was subjected to electrophoresis for different durations (0.5–15 min), while X represented the electrophoresis duration (in minutes). The hydrogel side connected to the Cu anode during electrophoresis was denoted as the + side, and the side connected to the graphite cathode was denoted as the − side.

### 4.3. Mechanical Properties and Swelling Behavior

The hydrogels were cut into dumbbell shapes (2 cm in narrow parallel width, 3.5 cm in length, and 2 mm in thickness), and the tensile strength of the hydrogel was measured in a Universal Testing Machine (CMT-1104, SUST, Xi’an, China) with a crosshead speed of 100 mm/min until the hydrogel broke. The force–displacement curve, the tensile strength (axial tensile force at the breakage point divided by the cross-sectional area of the hydrogel, i.e., 0.4 cm^2^), and the elastic modulus (the slope in the linear range of the stress–strain curve, i.e., 0% to 2% tensile strain in the stress–strain curve) of the hydrogel were recorded. In the case of the compression test, the hydrogel was cut into discs (10 mm in diameter and 2 mm in thickness), loaded onto the Universal Testing Machine, and compressed at a strain rate of 10% per minute until 90% strain. The compressive strength at 90% compression strain was recorded. In a cyclic compression test, the hydrogel disc was compressed to 50% strain and then relieved at a crosshead rate of 10% strain per minute. The highest compressive stress is recorded in each cycle.

To assess the swelling properties, the hydrogel was cut into discs (10 mm in diameter and 2 mm in thickness) and immersed in 15 mL of deionized water or artificial sweat at 37 °C for a given period. Excess water was removed from the surface of the swollen hydrogel using filter paper. The hydrogels were weighed before and after swelling and the swelling rate of the hydrogels was calculated using Equation (1):(1)$$S=\frac{W_{1}-W_{0}}{W_{0}}\times 100\%,$$
where *S* is the swelling ratio; and *W*_0_ and *W*_1_ are the weight of the hydrogels before and after swelling, respectively. After equilibration in deionized water and artificial sweat for 24 h, the hydrogels were frozen in liquid nitrogen, freeze-dried, and fractured to obtain cross-sections. The morphology of the hydrogel cross-sections was viewed using SEM with a voltage of 10 kV and a working distance of approximately 13 mm.

### 4.4. Determination of Cu Content and Cu Release from Hydrogel

The TAPS-Cu15 hydrogel discs (10 mm in diameter, 2 mm in thickness) were sectioned using a frozen slicer (NX500, Thermo, Waltham, MA, USA) into slices with a thickness of 0.2 mm. Slices from the + section (0–0.2 mm from the + side), the middle section (1–1.2 mm from the + side), and the − section (1.8–2 mm from the + side) of the hydrogel were collected and dissolved in nitric acid. The amount of copper from the slides was measured by ICP-OES.

For the release experiment, the TAPS-Cu15 hydrogel disk (10 mm in diameter, 2 mm in thickness) was placed on 4 mL of solidified agar gel (20 mm in diameter) in a 12-well plate, in which the + side or the − side of the hydrogel contacting the agar gel. The hydrogel with the agar gel was incubated at 37 °C for 5 days, during which the hydrogel was transferred to a new agar gel daily. The collected agar gel was dissolved in a nitric acid solution (2%), and the content of released Cu was measured by ICP-OES. The amount of Cu released from the hydrogel was calculated according to the standard curve. The cumulative release of Cu was calculated by adding the quantity of Cu released from the hydrogel every day.

### 4.5. Antibacterial Test

The bacteria were incubated overnight in the appropriate culture broth (tryptic soy broth for *S. aureus* and lysogeny broth for *E. coli*) and shaken at 100 rpm at 37 °C. The bacterial culture was diluted 1000 times with sterilized phosphate-buffered saline (PBS, 10 mM, pH 7.2) to obtain cell suspension (bacterial count 10^5^ CFU/mL). Hydrogels (10 mm in diameter, 2 mm in thickness) were placed in 5 mL of the bacterial suspension and cultured with shaking at 100 rpm at 37 °C. A suspension of PBS without hydrogels was used as the control group. The number of viable cells in suspension was counted by plate counting after 6 h of incubation.

### 4.6. Cytotoxicity Assay

Cytotoxicity tests were performed using NIH/3T3 fibroblasts according to the ISO-10993-5 standard with minor modification [51]. Cells in Dulbecco’s modified Eagle’s medium (DMEM, Hyclone, containing 10% fetal bovine serum, 100 mg/L streptomycin, and 1.0 × 10^5^ U/L penicillin) were added in a 96-well plate at a concentration of 10^4^ cells per well and cultured in DMEM at 37 °C and 5% CO_2_ for 24 h. Meanwhile, the hydrogels (500 mg) were disinfected under UV light for 30 min and incubated in sterilized deionized water (8 mL) at 37 °C for 24 h. The resulting extract solution was merged with DMEM at a volume ratio of 1:9. The mixture was used to culture the cells. The cells in the control groups were treated with a mixed deionized water/DMEM solution (1:9 volume ratio), fresh medium, and medium containing zinc diethyldithiocarbamate (10 mg/mL), respectively. After 24 h, cell viability was assessed using the cell counting Kit-8 (CCK-8, TransGen Biotech, Beijing, China) assay according to the manufacturer’s protocol. The percentage of absorbance value of the experimental group relative to the negative control group (deionized water/DMEM mixture) was expressed as cell viability.

### 4.7. Statistical Analysis

All data were presented as the mean ± standard deviation. The difference among different groups was compared, and statistically significant differences were marked with * (*p* < 0.05), ** (*p* < 0.01), *** (*p* < 0.001).

## Data Availability

All data and materials are available on request from the corresponding author. The data are not publicly available due to ongoing researches using a part of the data.

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
