# Peer review of "Biomimetic Gradient Hydrogels with High Toughness and Antibacterial Properties"

_gels, 2023, doi:10.3390/gels10010006_

Round 1

Reviewer 1 Report

Comments and Suggestions for Authors

Gels

Manuscript ID: Gels-2739050

Title: Biomimetic Gradient Hydrogels with High Toughness and Antibacterial Properties

In this study, the authors have hydrogels with both toughness and controlled drug release performance. The results seem to have potential for wound dressing application. However, the author can improve the manuscript by adding the recommended test method and modifying the manuscript as suggested.

Abstract

  1. TAPs-Cu abbreviation has not been presented before in the abstract; therefore, for the first time, the full word should be used before the abbreviation.

  2. The important results in terms of numbers, such as antibacterial activity and mechanical properties, should be presented in the abstract.   

  3. The authors wrote that “…. activity against gram-positive and gram-negative…..”. This indicates to the reader that the product is effective against every type of gram-positive and gram-negative microorganisms. The authors are required to specify the gram-positive and gram-negative bacterial strains utilized in the experiments.

Introduction

  1. “For example, using a freezing-assisted…….”. All tough hydrogel preparation methods should be indicated in the references.

Results and discussions

  1. What factors led the authors to conclude that Cu and the polymer were crosslinked? Further FTIR analysis is necessary to validate this explanation.

  2. In scheme 1, a scale bar of 5 mm in the caption should move to the bar in the scheme.

  3. There is no discernible distinction in Figure 1a. The authors may choose to magnify the region in which the distinctions are apparent.

  4. The meaning of the * symbols in Figures 1b, 2a, and 4 ought to be specified in the figure caption.

  5. The figure caption covers Figure 2c. Please revise.

  6. Why did the authors conduct a swelling test using artificial sweat? Does it accurately depict the practical application of this product?

  7. Can SEM really explain the crosslinking phenomena of the polymer? If the answer is yes, please add a reference.

  8. Please add the scale bar number in Figure 3.

  9. The Cupper distribution in the gel is not homogenous. Will this affect therapeutic efficiency?

  10. From Fig. 4c, the drug release seems to be constant from the first day, which cannot be seen in the actual drug release mechanism. I suggest the author examine the drug release at 5–10 points on the first day.

Materials and methods

  1. It is advisable to compare the mechanical properties of the gel that has been prepared with those of known gels, including HPMC and other well-known polymer gels. This may facilitate the reader's comprehension of the outcome.

Comments on the Quality of English Language

-

Reviewer 2 Report

Comments and Suggestions for Authors

Dear Authors,

Thank you for your manuscript.

The paper provides interesting data but still needs considerable revision to be acceptable for Gels journal.

Comments are listed as follows.

Comment 1.

I do not need an illustration of hydrogen and chelate bonds in Scheme 1. I looked in the diagram but could not find it (page 4)

Comment 2

If there are electrophoresis for different durations (0.5-15 min) (page 12, 4.2 section) or other experimental conditions, a table summarizing them in the text should be added. Please describe what it refers to, such as TAP-Cu0.5, so the reader can easily understand. This will be useful when we look at Figure 1-6.

Comment 3

Why is the cell survival rate over 100%? You can use other measurement methods if the authors want to use it as a survival rate. When using the Cell Counting Kit-8 assay, displaying this data differently on the vertical axis is desirable.

Best Regards

Round 2

Reviewer 1 Report

Comments and Suggestions for Authors

The manuscript is ready to be published.

Reviewer 2 Report

Comments and Suggestions for Authors

Dear Authors,

Thank you for your manuscripts.

This paper is a significant contribution, and I recommend it be accepted for publication.